# Phylogenetic rewiring in mycorrhizal–plant interaction networks increases community stability in naturally fragmented landscapes

Alicia Montesinos-Navarro [1]*, Gisela Díaz[2], Pilar Torres[2], Fuensanta Caravaca[3] & Antonio Roldán[3]

Although ecological networks are usually considered a static representation of species' interactions, the interactions can change when the preferred partners are absent (rewiring). In mutualistic networks, rewiring with non-preferred partners can palliate extinction cascades, contributing to communities' stability. In spite of its significance, whether general patterns can shape the rewiring of ecological interactions remains poorly understood. Here, we show a phylogenetic constraint in the rewiring of mycorrhizal networks, so that rewired interactions (i.e., with non-preferred hosts) tend to involve close relatives of preferred hosts. Despite this constraint, rewiring increases the robustness of the fungal community to the simulated loss of their host species. We identify preferred and non-preferred hosts based on the probability that, when the two partners co-occur, they actually interact. Understanding general patterns in the rewiring of interactions can improve our predictions of community responses to interactions' loss, which influences how global changes will affect ecosystem stability.

[1] Centro de Investigaciones Sobre Desertificación (CIDE, CSIC-UV-GV), Carretera de Moncada-Náquera Km 4.5, 46113 Moncada, Valencia, Spain. [2] Departamento de Biología Aplicada, Área de Botánica, Universidad Miguel Hernández, Elche, Alicante, Spain. [3] CSIC—Centro de Edafología y Biología Aplicada del Segura, Department of Soil and Water Conservation, Campus de Espinardo, Murcia, Spain. *email: ali.montesinos@gmail.com

The anticipation of how biodiversity shifts will affect the robustness of ecological communities is a major goal in the assessment of the risks of global changes. The first attempts to predict co-extinction cascades focused on the architecture of ecological networks, considering the topology of the ecological interactions as a static entity[1,2]. However, there is growing interest in improving the realism of theoretical co-extinction cascades by considering the topological dynamics of ecological networks[3–5]. This modern theoretical framework introduces the concept that species can rewire with other partners when the preferred one is absent, resulting in a different network topology that in turn influences the fate of the species. This is the case of non-native species that colonize a new habitat and establish interactions with species absent in their original range of distribution[6,7]. Shifts in the topology of ecological networks are likely to occur in fragmented landscapes where, while still harboring similar communities, the variability in fragments area might slightly shape species composition. Despite the increasing attention that rewiring has received, we still lack a deep understanding of its contribution to the diversity of interactions across communities, the potential rules governing the rewiring of interactions, or its implications for the stability of natural communities.

Most ecological interactions are influenced by species traits, which can be to some extent evolutionary conserved[8]. Based on this reasoning, it has been shown that ecological interactions are evolutionarily conserved across the tree of life[9], which provides the basis to hypothesize that interactions with non-preferred partners might not be freely established, and rewiring could be phylogenetically constrained[10] (i.e., when a preferred partner is absent, the establishment of interactions with non-preferred partners might be prone to occur with close relatives of those preferred). The contribution of a phylogenetically constrained rewiring to enhance the community robustness is uncertain. On the one hand, rewiring in mutualistic networks can increase the community robustness to face species loss, slowing down extinction cascades[4,11], although the opposite can be true in trophic networks - where rewiring can result in overexploitation of the resources[12]. On the other hand, a phylogenetically constrained rewiring (or a constraint of any other nature) could also reduce the enhancement of the community robustness compared to a completely free rewiring. The quantification of the balance between these opposing forces can improve our understanding of how ecological networks overcome disturbances resulting in species loss. However, whether rewiring can be phylogenetically constrained, and its implications for community stability, is largely unknown.

Most vascular plant species associate with arbuscular mycorrhizal fungi forming the most ancient symbioses on Earth[13]. Arbuscular mycorrhizal fungi are obligate symbionts that depend on their plant hosts to complete their life cycle[13], so that the absence of all their hosts might likely result in the fungal extinction from the community. On the contrary, plants can survive without the mycorrhizal fungi, being less likely to disappear due to the loss of all their mycorrhizal partners. Although mycorrhizal symbiosis has been considered highly generalist[14], recent evidence has shown that in natural communities plant-mycorrhizal fungi interactions are, to some extent, species-specific (i.e., non-random)[15], suggesting that fungal partners might show preferred and non-preferred hosts among the plant species present in a community. Naturally fragmented habitats, such as gypsum soils immersed within other lithologies, harbor relatively similar plant communities[16], with some species turnover across fragments. A set of networks with similar nodes (i.e fragments with similar species composition), but slightly different interaction patterns, can maximize the probability of finding multiple pairs of co-occurring species that might (or might not) interact in each fragment. Shifts in the mycorrhizal interactions across gypsum fragments might allow estimating how likely a given interaction is to occur (or not) (i.e., frequency) between two species across fragments, as long as they co-occur. The interactions that are likely to occur, whenever the two partners co-occur, can be assumed to involve preferred hosts, while the interactions that are unlikely to occur, even when both partners co-occur in the same fragment, can be considered to involve non-preferred hosts. Therefore, the comparison of multiple mycorrhizal networks across fragmented gypsum landscapes provides a suitable system to elucidate patterns of interactions' rewiring in ecological networks.

In this study, we selected 15 fragments of gypsum outcrops, ranging from less than one to 85 ha, to assess rewiring patterns in the mycorrhizal symbiosis. We hypothesize that networks rewiring can be phylogenetically constrained, so that interactions with non-preferred hosts tend to occur with close relatives of the preferred hosts. In addition, we assess the effect of a phylogenetically constrained rewiring on the robustness of these mutualistic networks under three simulated scenarios: no rewiring ("noRW"), phylogenetically constrained rewiring ("RWphylo"), and free rewiring ("RWrand"). Our results show that the non-preferred hosts of arbuscular mycorrhizal fungi are not randomly selected from the potential hosts present in each fragment. Instead, the non-preferred hosts that actually harbor a fungi tend to be more closely related to (i.e., show a minimum phylogenetic distance to) a preferred host than expected by chance. This phylogenetic constraint in interactions' rewiring has implications for the robustness of fungal communities. When the output of the three simulated scenarios are compared, the robustness of fungal communities to plant species loss increases when rewiring is allowed. Even when rewiring is phylogenetically constrained, the robustness increases 64.5% of the potential increase without any constraint. As far as we know, this is the first study that, through the assessment of shifts in the mycorrhizal symbiosis patterns across gypsum fragments, describes a phylogenetic constraint in the rewiring of mutualistic interactions, providing the arena for further research on whether this can be a general pattern for other mutualistic networks. Realistic assessments of rewiring patterns in ecological interactions are necessary to accurately predict the effect of future changes on ecosystem stability.

## Results

**Study overview**. We sampled 15 individuals along a random transect, which length was proportional to each fragment's size (i.e., 225 individuals across fragments). Across the 15 networks, the individuals sampled represent the relative abundances of the plant species in each fragment. We extracted DNA from the roots of each plant individual by using the MOBIO Power Soil® DNA Isolation kit (MO BIO Laboratories, Inc., Carlsbad, CA, USA). We used Illumina MiSeq platform 2 × 300 bp and an amplicon-based and tagmentation approach (i.e., universal tags incorporated into DNA fragments) to sequence the 18 s rDNA region. After grouping fungal sequences into operational taxonomic units (OTU), we assessed the plant-fungal interaction network in each fragment, and characterized network rewiring (i.e., changes in the configuration of the networks through shifts in their links) by considering that an interaction was the result of a rewiring event if it was established with a non-preferred host. To classify hosts into preferred and non-preferred, we distinguished between interactions that were realized in most of the fragments in which the two partners were present (non-rewired interactions, involving preferred host), and those interactions that were not usually realized despite the two partners were present (rewired

interactions, with non-preferred hosts). We recorded 2494 different mycorrhizal interactions involving 213 plants, of 28 species and 13 families, and 161 arbuscular mycorrhizal OTUs, of seven families (Supplementary Table 1).

We use this data to test whether networks rewiring can be phylogenetically constrained, so that interactions with non-preferred hosts tend to occur with plant species that are close relatives of the preferred hosts. Then, we assessed to which extent a phylogenetic constraint in the interactions' rewiring affects the robustness of these networks. Robustness is defined as the area below the extinction curve described by the number of fungal OTUs that disappear in each sequential host simulated removal, assuming that fungal OTUs will be extinct when all their hosts had been removed from the community.

**Phylogenetically constrained rewiring.** In order to test our hypothesis, for each OTU in each fragment we calculated the averaged phylogenetic distance between the non-preferred hosts harboring the OTU, and their closest relatives among the OTU's preferred hosts across fragments. The observed mean phylogenetic distance is compared with that expected under a simulated null model in which the non-preferred hosts harboring the OTU are randomly selected from the set of plant species present in the fragment, excluding the preferred plant species. A lower phylogenetic distance observed compared to that expected under the null model, will suggest that rewiring tends to occur with plant species that are close relatives of the preferred hosts (Fig. 1). From the 1623 presences of the 161 OTUs across the 15 fragments, we only used in this analysis OTUs that co-occur with the same plant species in 4 or more fragments, and showed preferred and non-preferred hosts (72 OTUs, 388 presences across fragments). Based on this data, our results support the hypothesis that network rewiring is phylogenetically constrained, as the average phylogenetic distance observed is significantly lower than that expected by the null model ($t = -2.02$, df = 387, $p$-value = 0.04, estimate of the difference observed-expected = $-6.57$) (Fig. 2).

**Network robustness.** In order to assess to which extent a phylogenetically constrained rewiring affects the robustness of the mycorrhizal networks, we used a simulation approach that compared the fungal robustness to plant species loss under three scenarios: in the absence rewiring ("noRW") (i.e the fungal OTUs present in the removed plant were not allowed to rewire to other hosts), allowing a phylogenetically constrained rewiring ("RWphylo") (i.e the fungal OTUs were allowed to rewire but only to hosts of the same family of the removed host), and allowing a free rewiring ("RWrand") (i.e the fungal OTUs was allowed to rewire to any host present in the fragment). In order to consider a realistic amount of rewiring in our simulations, we firstly calculated the contribution of rewiring and species turnover to the shifts in interaction patterns across fragments (sensu Poisot et al.[17]). This approach decouples the dissimilarity in interaction patterns between every pair of fragments into two components: their dissimilarity in species composition (i.e., species turnover $\beta_{ST}$) and their differences due to a different pattern of interaction between the species shared by both networks (i.e., network rewiring $\beta_{WN}$). The contribution of species turnover relative to rewiring (i.e., the ratio $\beta_{ST}/\beta_{WN}$), averaged across all paired comparisons, was (mean ± SE) 64 ± 1.1% (Supplementary Table 2), showing that rewiring plays a relevant role explaining dissimilarities in interactions patterns across fragments.

Finally, we used this estimated contribution of species turnover relative to rewiring to establish realistic probabilities of rewiring in our simulated extinction cascades. In all the simulated extinction cascades we started from a matrix representing the presence of fungal OTUs (rows) within the roots of plant individuals (columns) in each fragment. In each step, one random plant individual was removed from the interaction matrix. Once the last partner of a specific fungal OTU was removed from the community, this obligate symbiont was assumed to be extinct. Network robustness ranges from 0 to 1 and represents the fraction of host species that need to be removed to result in a greater than 50% total loss of the fungal OTUs.

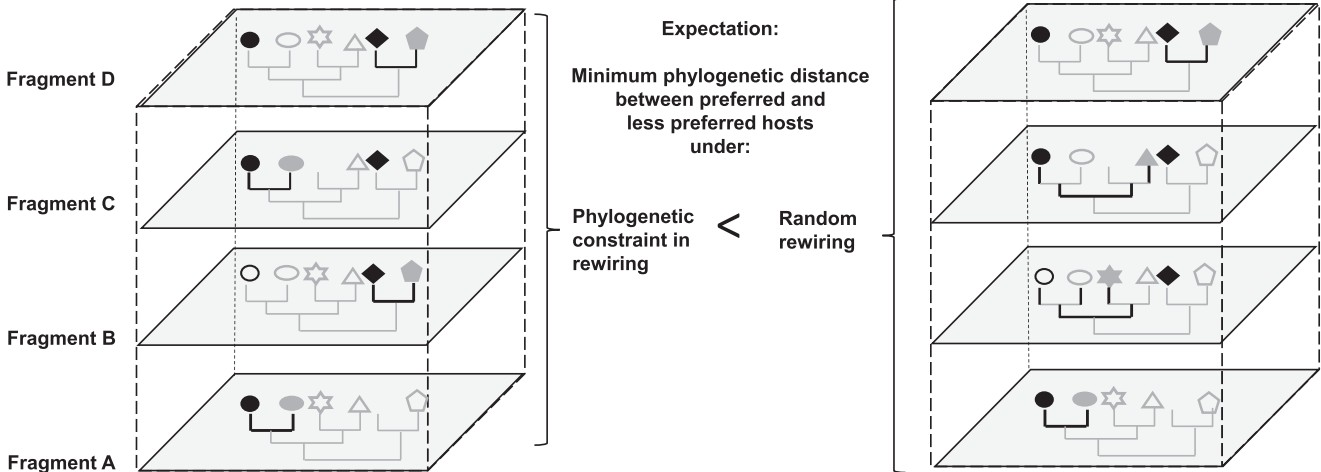

**Fig. 1 Conceptual representation of the approach used to assess a phylogenetic constraint in the rewiring of interactions.** Different shapes represent every plant species with which a given fungal species has interacted across all the fragments. Within the subset of potential hosts occurring in each fragment (large rectangles) it is distinguished which plant species actually harbor the fungal species (filled shapes) or not (empty shapes) in that specific fragment. Hosts that harbor the fungal species in more than half of the times that both partners occur in the same fragment are considered preferred hosts (in black), while the rest are considered non-preferred hosts (in gray). In the presence of a phylogenetic constraint, it is expected that those hosts harboring a given fungal species despite being a non-preferred host (i.e., rewiring), will tend to be close relatives of the preferred hosts, thus, showing a low minimum phylogenetic distance between them. In the absence of the phylogenetic constraint, the non-preferred hosts that actually harbor the fungal species are expected to be randomly distributed in the plant phylogeny, resulting in an increase in the - averaged across fragments- minimum phylogenetic distance between preferred and non-preferred hosts The minimum phylogenetic distance is highlighted using black lines.

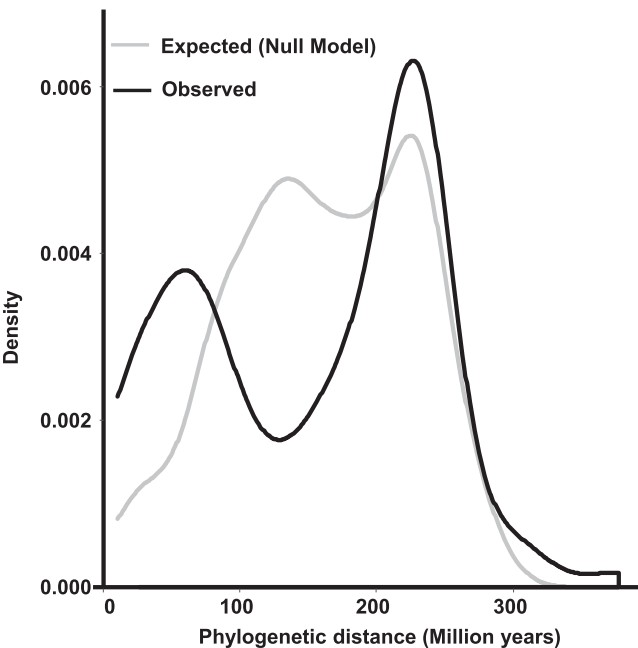

**Fig. 2 Density plot comparing the observed and expected distribution of the minimum phylogenetic distances between preferred and non-preferred host species.** Fungal DNA was extracted of the roots of each plant, grouped into operational taxonomic units (OTU), and used to assess plant-fungal interaction networks. For each fungal operational taxonomic unit (OTU), the information across the 15 fragments was used to define its preferred (or non-preferred) hosts. Preferred hosts were considered those plant species that actually harbored the fungal OTU in more than half of the times that both partners occur in the same fragment (i.e., number of opportunities to interact). For each OTU in each fragment ($n = 388$), it was calculated the average phylogenetic distance from its non-preferred hosts (but actually harboring the OTU) to their nearest relative among the preferred hosts (black line). Simultaneously, for each OTU in each fragment, the same number of hosts was randomly selected from the set of plant species present in the fragment, excluding the preferred plant species (gray line).

We simulated extinction cascades under three scenarios: in the absence rewiring ("noRW"), allowing a phylogenetically constrained rewiring ("RWphylo"), and allowing a free rewiring ("RWrand"). In the scenarios allowing rewiring, in each step, rewiring was actually realized or not with a probability equal to 1 - the estimated $\beta_{ST}$/$\beta_{WN}$. Across the 15 fragments, the robustness of the fungal communities to the loss of plant species was higher under the "RWrand" scenario than under the "noRW" scenario, and the estimates regarding the "RWphylo" scenario were always in between the two (Fig. 3). Rewiring increased the network robustness, and when it was phylogenetically constrained, the mean increase (RWphylo-noRW) was (mean ± SE) 64.5 ± 3.9% of the potential increase in robustness due to rewiring without any constraint (RWrand-noRW) (Supplementary Table 3).

## Discussion
The mycorrhizal networks shifted across fragments. The contribution of species turnover (relative to rewiring) ranged from 29% to 84%, implying that rewiring in mycorrhizal symbiosis was relatively frequent across sites. However, certain constraints seemed to apply to the establishment of interactions with non-preferred hosts when the preferred hosts of the obligate fungal symbionts were absent. Mycorrhizal fungi tended to interact with the non-preferred hosts that were close relatives of their

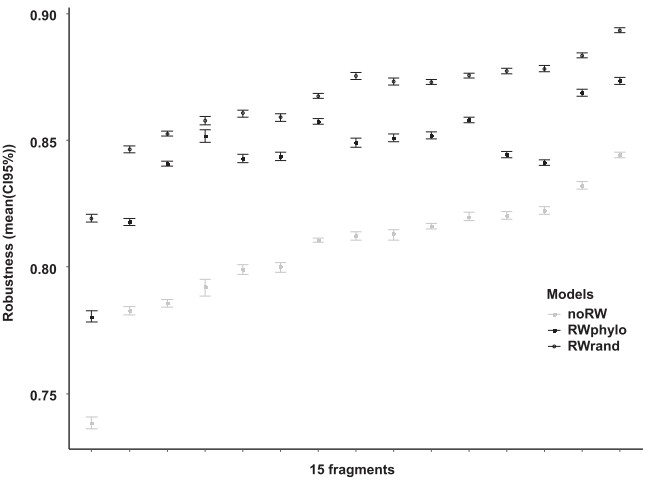

**Fig. 3 Fungal community robustness to the loss of plant partners (mean and 95% confidence intervals).** Robustness represents the fraction of host species that need to be removed to result in the loss of >50% of the fungal OTUs in a community. A specific fungal OTU is extinct when all its hosts have been removed. Fungal OTUs and its host plant species are defined based on the fungal DNA extracted of the roots of each plant individual. The robustness was estimated under three scenarios: without allowing rewiring (noRW), allowing random rewiring without any constraint (RWrand), and allowing phylogenetically constrained rewiring (RWphylo) within plants of the same family as the original fungal partner. One hundred simulations ($n = 100$) of the three scenarios in each of the 15 fragments are presented ordered by increasing robustness of their networks under the noRW scenario.

preferred hosts, which could limit the benefits of rewiring for fungal community stability. A potential mechanism underlying this pattern might be mediated by mycorrhizal fungi differing in their hyphae tendency to colonize root vs. soil[18], which can influence host preference based on the plant root structure. Many plant root traits such as root diameter, root tissue density, and root nitrogen content are phylogenetically conserved in seed plants[19], potentially enhancing a mycorrhizal fungal preference for similar hosts. Nevertheless, these results should be interpreted with caution as due to the difficulty to assess networks rewiring in field conditions, our results are based on an approach of space-by-time substitution, using snapshot of many fragments in the region. Ideally, host preference could be assessed experimentally, exposing fungi to every potential host under similar conditions, and quantifying the fungal association with each plant species relative to the rest, and even monitoring this process over time. However, many of the fungal species found in this study might be difficult to cultivate, what prevents using them in experimental designs, and furthermore, conducting these experiments in a broad range of plant species under field conditions might easily become unfeasible. Also, the phylogenetic distances between the plant species have not been established by genotyping plant material, and therefore the estimates of the phylogenetic distances between species might be slightly unaccurate, although we have used the well established relationships at the family level. However, considering that many of the families used are monospecific, and the highest number of species within a family is low (i.e., 4), our approach is likely to provide an acceptable estimate of the phylogenetic distances between the plant species used in this study. Thus, considering the interest and difficulty to assess rewiring patterns in the wild, our attempt provides valuable information that could guide further necessary investigation.

This study quantifies the contribution of rewiring to explain the shifts of mycorrhizal networks across fragments, and elucidate a phylogenetic pattern in the interactions rewiring (i.e., rewiring tends to occur with close relatives of the preferred hosts). Besides, we simulate the implications of a phylogenetically constrained rewiring on community stability, showing that, despite this phylogenetic constraint, rewiring increases the robustness of the fungal communities to the loss of their hosts. Although free rewiring would increase even more the fungal community robustness, a phylogenetically constrained rewiring still achieves, overall, 64.5% of the increase that could be achieved with free rewiring. Many relevant traits shaping species interactions are phylogenetically conserved[9], suggesting that a phylogenetically constrained rewiring could be widespread across a broad diversity of ecological interactions. Our results deepen the understanding of the potential rules governing the rewiring of interactions, and will contribute to improve future predictions of the effects of global change on the extinction cascades that currently threat biodiversity.

## Methods

**Study design**. Our study system was a naturally fragmented gypsum area composed by 15 lithological fragments (0.15–85 ha) located in Southeastern Spain (37° 40′N, 1°41′W) (Supplementary Fig. 1). The climate is semiarid Mediterranean, with a mean annual temperature of 16 °C, average annual rainfall of 289 mm, mainly concentrated in early spring and fall, and a pronounced drought season in summer. Gypsum ecosystems are very restrictive environments which occur in arid and semiarid regions and are subject to particularly stressful conditions due to the properties of the soil and climate. In the study system, these ecosystems usually have a fragmented spatial structure because the soil has a natural patchy distribution, as a mosaic of edaphic islands immersed within other lithologies or geological substrates mainly limestone. The soils are classified as Petrogypsic Gypsiorthid (SSS, 1999) and Gypsisol (FAO, 1998), with a gypsic and petrogypsic horizon within 100 m of the surface developed on gypsum parental rocks. The vegetation is an open (40% vegetation coverage) scrubland with woody gypsophilous (i.e., that preferentially or only establish in gypsum soils) perennial species, and other plant species that do not exclusively grow in gypsum soils. The plant community is rich in germanders (*Teucrium*), thymes (*Thymus*), rockroses (*Helianthemum*), ruptureworts (*Herniaria*), composites, and grasses.

We used high-resolution aerial photographs (http://www.ign.es) to identify remnant patches of gypsum soil and vegetation, which were identified and georeferenced in the field. We selected 15 outcrops of gypsum soil (i.e., fragments) ranging between 0.15 and 85 ha (Supplementary Table 1). The area of each fragment was measured in ArcGis 10.5 (ESRI, Redlands, CA, USA).

**Field sampling**. In order to have representative samples of the plant communities, we estimated the plant species composition and abundance in each fragment using line-transects, which were proportional in length to the area of the fragment. Therefore, despite the variation in fragment size, the sampling ensured a similar proportion of sample coverage across fragments. The number of individuals for each plant species was recorded and the relative abundance of each plant species was determined. Based on the relative abundances of the plant species in each fragment, 15 plant individuals per fragment were selected for sampling (Supplementary Table 1).

Plant individuals were collected in May 2015 (late spring) along a line-transect in the core of each fragment, to avoid major edge effects. We sampled 15 individuals per fragment for a total of 225 individuals belonging to 28 plant species. They were placed in polyethylene bags and transported to the laboratory. Roots were separated from shoots, washed, and dried with filter paper. In total 500 mg of roots from each plant were cut into 1 cm pieces, carefully mixed, and stored in vials at −18 °C for molecular analyses.

Next Generation Sequencing (NGS) approaches are used as a tool to study arbuscular mycorrhizal fungi communities as they provide sufficient depth to deliver insights into variations in environmental samples. These approaches allow infrequent fungi to be detected and increase the proportions of diversity captured[20]. The Illumina sequencing platform is a particularly useful approach because of its low error rate and high sequencing depth per sample. We used the Illumina MiSeq platform 2 × 300 bp, which produces longer DNA reads than the previous 2 × 200 bp length platform, and an amplicon-based and tagmentation approach (i.e., universal tags are incorporated into DNA fragments) in order to allow longer DNA regions, such as the 18 s rDNA region, to be sequenced.

Total DNA was extracted from 500 mg of roots of each plant individual (sample) by using the MOBIO Power Soil® DNA Isolation kit (MO BIO Laboratories, Inc., Carlsbad, CA, USA). After the procedure, the concentrations of DNA in the samples were measured using an Appliskan fluorescence-based

microplate reader (Thermo Scientific Wilmington, US). Amplicon library preparation for the Illumina MiSeq platform was based on the Illumina MiSeq system preparation manual "16S Metagenomic Sequencing Library Preparation". The region-specific primers NS31 and AML2[21,22] were used to target the 18s rDNA V4 region of arbuscular mycorrhizal fungi, with overhang adapters attached.

Thus, the composite primers were forward 5′ TCGTCGGCAGCGTCAGATGTGTATAAGAGACAGTTGGAGGGCAAGTCT GGTGCC and reverse 5′ GTCTCGTGGGCTCGGAGATGTGTATAAGAGAC AGGAACCCAAACACTTTGGTTTCC. For Index PCR, Illumina Nextera XT v2 Index Kit primers were used. The forward index primer was: AATGATA CGGCGACCACCGAGATCTACAC[i5]TCGTCGGCAGCGTC; and the reverse index primer was: CAAGCAGAAGACGGCATACGAGAT[i7]GTCT CGTGGGCTCGG, where i5 and i7 indicate the position of forward and reverse index sequences.

The PCR was carried out in two sequential reactions: Amplicon PCR with region-specific primers and partial adapters for sequencing; and Index PCR, which was used to complete the sequencing adapters and add Nextera XT sample identifying indexes. Between sequential reactions, the amplicons were purified with AMPure bead technology. In the Amplicon PCR 3 μL of DNA sample were used, while 3 μL of purified amplicon were used in the Index PCR reaction.

The total reaction volume was 30 μL. The PCR reactions contained primers (0.2 μM, final concentrations) and Smart-Taq Hot Red 2× PCR Mix (0.1 U/μL Smart Taq Hot Red Thermostable DNA Polymerase, 4 mM MgCl₂, 0.4 mM dATP, 0.4 mM dCTP, 0.4 mM dGTP, 0.4 mM dTTP; Naxo OÜ, Estonia). An NTC (no template control) was added to all PCR reactions. A 1-kb DNA Ladder (Naxo OÜ, Estonia) was used as a size marker on all electrophoresis gels.

Amplicons of the 18 s DNA V4 region produced by Index PCR were purified with an Agencourt AMPureXP Kit (Beckman Coulter Inc. Brea, US). Samples were eluted in Buffer EB (10 mM Tris-HCl, pH 8.5; QIAGEN Inc). The DNA concentrations of the purified amplicons were measured using an Appliskan fluorescence-based microplate reader (Thermo Scientific) and PicoGreen® dsDNA Quantitation Reagent (Quant-iT dsDNA Broad Range Assay Kit, Invitrogen Wilmington, US). After measurement of the concentrations of the amplicons, those with a concentration of at least 10 ng/μL were pooled equimolarly (200 ng of each sample). The concentration of DNA in the pool was measured in a QubitTM fluorometer, in three replicates, resulting in a mean value of 43.9 ng/μL. In total 12 samples failed to yield a detectable amplicon in the Index PCR and thus were excluded from the sequencing library.

The DNA extraction and the preparatory procedures for sequencing were performed using BIOTAP LLC (Tallinn, Estonia). The sequencing of the DNA library was performed on the Illumina MiSeq 2 × 300 bp v3 run platform by Microsynth AG (Balgach, Switzerland).

**Bioinformatics**. Demultiplexing and Illumina adaptor trimming were performed, and Row R1 and R2 Miseq reads were obtained in FASTQ format. The quality of the obtained reads was checked with the software FASTQC (http://www. bioinformatics.babraham.ac.uk/projects/fastqc/). The processing of the FASTQ files, containing the reads, was carried out with Qiime 1.9.0[23], which performs standard microbial community analyses, including quality filtering of reads, operational taxonomic unit (OTU) picking, and taxonomic assignment.

After quality filtering of the reads (minimum Phred quality score of 20), chimeras were detected and removed using the UCHIME algorithm implemented in VSEARCH[24]. The remaining filtered and cleaned 18 S reads were then clustered into OTUs through the closed-reference approach in Qiime, using the MaarjAM database as the reference[25]. Eleven sequences of glomeraceae out of the 1799 unique sequences did not match with any of the virtual taxa in the MaarjAM database and were not used in further analyses. Each OTU was assigned to fungal taxa using the UCLUST algorithm[26]. Data on the OTU abundance in each sample at different taxonomic levels were obtained. Then, a second quality filtering was carried out: we removed the OTUs with a number of sequences lower than 0.005% of the total number of sequences. Finally, the rarefaction plots were constructed, to show the rarefied number of OTUs defined at a 97% sequence similarity threshold. When the rarefaction curves tended towards saturation, the sequencing depth was assumed to be sufficient to retrieve most of the arbuscular mycorrhizal diversity. In addition, the percentage of coverage was calculated by Good's method[27]. The bioinformatic analyses were performed by All Genetics & Biology, SL, A Coruña, Spain. The accumulated number of OTUs per plant sampled in each fragment is presented in Fig. 4.

**Statistics and reproducibility**. Contributions of rewiring and species turnover: Shifts in interaction's patterns between two sites (i.e., network dissimilarity) can be due to two components, species turnover (i.e., the species present differ between fragments) and network rewiring (i.e., the same species are present but the interaction pattern among them differs between fragments). We estimated the relative contributions of these two components based on Poisot et al.[28], who proposed that the overall differences in interaction patterns between two networks ($\beta_{WN}$) result from the combination of differences in the interaction structure due to dissimilarity in species composition ($\beta_{ST}$) (i.e., species turnover) and differences due to a different pattern of interaction between the species shared by both networks ($\beta_{OS}$) (i.e., network rewiring). Two interesting conclusions can be extracted from this additive partitioning of the

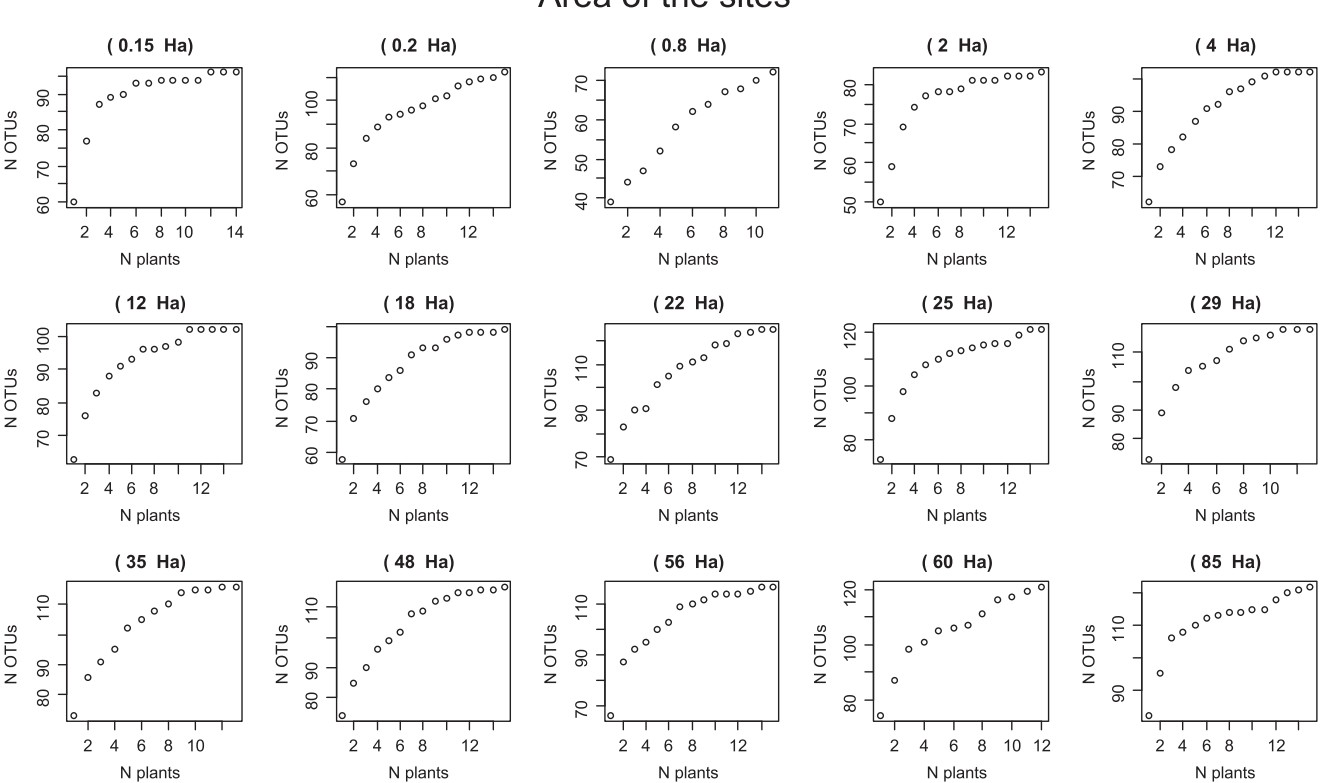

**Fig. 4** Accumulated number of fungal operational taxonomic units (N OTUs) per plant individual sampled in each of the 15 fragments.

sources of dissimilarities between interaction's patterns: while $\beta_{WN}$ and $\beta_{ST}$ co-vary with the species composition dissimilarity between networks, $\beta_{OS}$ does not; And, given that $\beta_{OS}$ is a component of $\beta_{WN}$, $\beta_{OS} \leq \beta_{WN}$, and thus $\beta_{ST}$ takes values between 0 (the overall dissimilarity is entirely explained by rewiring) and $\beta_{WN}$ (there is no rewiring and all the differences in interaction patterns are due to species turnover). Therefore, the ratio $\beta_{ST}/\beta_{WN}$ provides a measurement of the contribution of species turnover (relative to rewiring) to the overall dissimilarity in the interaction patterns between the two networks compared.

We calculated the different components of the dissimilarity in the interaction patterns ($\beta_{WN}$, $\beta_{ST}$, $\beta_{OS}$, and $\beta_{ST}/\beta_{WN}$) across every possible pair of the 15 fragments ($N = 105$ pairs of fragments) using the "network_betadiversity" function in the "betalink" packages of R v. 3.2.2[17]. The estimates of $\beta_{ST}/\beta_{WN}$ were used to establish realistic probabilities of rewiring in the following simulations to assess network robustness.

**Phylogenetically constrained rewiring**. Using the information across the 15 fragments, we quantified how many times each pair of plant species-fungal OTUs occurred in the same fragment, and considered it as the number of potential interactions between them. Then, for each fungal OTU we defined the preferred hosts as the plant species that actually harbored it in >50% of their potential interactions (i.e., times that both partners co-occur across fragments). We took into account only cases in which the number of potential interactions were 4 or more, avoiding extremely high or low stochastic percentages due to a low number of potential interactions. The rest of plant species that also harbored the fungal OTUs were considered non-preferred hosts.

In order to calculate the phylogenetic distances between non-preferred and preferred hosts we first generated the phylogenetic relationships among all the plant species sampled with the R function S.PHYLOMAKER[29]. This function uses the PhytoPhylo backbone megaphylogeny, which is an updated version of the time-calibrated angiosperm species-level phylogeny[30]. The community phylogeny was produced by matching the family names of the plant species sampled in the studied fragments with those in the backbone phylogeny, using the R package APE[31]. Finally, the phylogenetic distances between non-preferred and preferred hosts were obtained using the 'cophenetic' function in the APE package[31].

For each OTU in each fragment ($N = 1623$ presences of the 161 OTUs across the 15 fragments) we calculated the minimum phylogenetic distance between the non-preferred hosts that actually harbored each OTU in that fragment, and their respective closest relative among the preferred hosts of each OTU, and then averaged the values for all the non-preferred hosts of each OTU in each fragment.

Simultaneously, we built a null model by randomly selecting a set of non-preferred hosts that could have potentially harbored each OTU in each fragment. To do so, from the plant species present in each fragment (excluding the preferred hosts for each OTU), we randomly selected the same number of observed non-preferred hosts harboring the OTU, and calculated the same metric described in the previous paragraph. Finally, we tested whether the difference between the observed and expected values, based on the null model, was significantly different from 0 using a $t$-test conducted with the t.test function in the base package of R v. 3.5.2[28]. The iterations to calculate the phylogenetic distance for the observed networks and the null model were performed using R v. 3.5.2[28] (Supplementary Code 1)[32].

**Network robustness**. We simulated extinction cascades in fungal communities due to the simulated loss of their plant hosts under three scenarios: without rewiring ("noRW"), allowing phylogenetically constrained rewiring ("RWphylo"), and allowing non-constrained rewiring ("RWrand"). In all cases, we started from the interaction matrix representing the presence of fungal OTUs (rows) within the roots of plant individuals (columns) in each fragment, which takes into account the relative abundance of plant species and the prevalence of fungal OTUs in the plant community. In each step, one random plant individual was removed from the interaction matrix. In the "noRW" scenario, the fungal OTUs present in the removed plant were not allowed to rewire to other hosts. Once the last partner of a specific fungal OTU was removed from the community, this obligate symbiont was assumed to be extinct. In the "RWphylo" scenario, the fungal OTU present in the plant individual removed in each step was allowed to rewire only to plants within the same family as the hosts in which it was observed, if they were present in the interaction matrix. A probability that the rewiring would eventually occur in each step was assigned based on the relative contribution of species turnover/rewiring estimated by following Poisot et al.[33] ($1-(\beta_{ST}/\beta_{WN})$, Supplementary Table 2). This relative contribution had been estimated for all possible pair-wise comparisons of the 15 networks, and one of those estimates was randomly selected in each step. Finally, the "RWrand" scenario was exactly the same as the previous one, except for the fact that fungal OTUs whose partners had been removed were allowed to rewire to any randomly chosen plant species still present in the interaction matrix. We used the "robustness" function in the "bipartite" package to estimate this network property[34]. This function simulates the removal of plant species and consequent loss of the obligate mutualistic fungi that depend upon them for reproduction. For each network, the plant species were removed sequentially, randomly selecting which would be the next species to remove. Fungal diversity declines slowly or rapidly depending on the structure of the interaction network. Therefore, the

robustness of the network quantifies the speed of this decline, and is measured as the area below the extinction curve described by the number of fungal OTUs that disappear in each sequential plant species removal[35]. Robustness was quantified with a single parameter R that ranges from 1 (i.e., a curve that decreases slowly until almost all hosts have been removed) to 0 (i.e., a curve that decreases abruptly when just a few hosts are lost)[36]. Each simulation was repeated 100 times to obtain the 95% confidence interval of the robustness estimate for each fragment (Supplementary Table 3). The simulations were performed using R v. 3.5.2[28].

**Reporting summary**. Further information on research design is available in the Nature Research Reporting Summary linked to this article.

## Data availability
The data used is available in the National Center for Biotechnology Information Search database (NCBI); Sequence Read Archive (SRA) accession: PRJNA516318. Other source data are available in Supplementary Data 1–3.

## Code availability
All analyses were performed in R software version 3.5.2. R Core Team, R: A language and environment for statistical computing. R Foundation for Statistical Computing, Vienna, Austria. 2013, (2015). The R code used in this study is provided as Supplementary Code 1.

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

## Acknowledgements

We are grateful to Dr David Walker for kindly correcting the English language. Financial support was provided by the Spanish National Research Program for Development and Innovation CGL2013-42312-R / BOS. AMN was supported by a postdoctoral contract from the Spanish Ministry of Economy and Competitiveness (FPDI-2013-16266; IJCI-2015-23498).

## Author contributions
All authors participated in the study conception and the design of the methodology. G.D., P.T. and A.R. conducted the data collection, AMN performed the computation and formal analyses and wrote the initial draft, and all authors contributed to the critical review of the manuscript. G.D., P.T., F.C. and A.R. contributed with funding acquisition

## Competing interests
The authors declare no competing interests.
