## [Peer Review File · Communications Biology]

Reviewers' comments:

Reviewer #1 (Remarks to the Author):

The manuscript is generally very well written and makes a nice original contribution to ecology. They sampled mycorrhizal fungal plant associations on the fragmented gypsum soil landscape, and compared network structures among fragments to characterize the effects of fragmentation on the mycorrhizal mutualistic networks using the approach of space-for-time substitution. Notably, this is a laudable attempt to address interaction rewiring on a fragmented landscape, a highly novel aspect of fragmentation effects on communities. However, after reading the manuscript a few times, I see a few major concerns about the statistical analysis and the conclusions they made.

First, despite strong focus on interaction network, the authors sometimes analyzed plant individual as an independent statistical unit for significance testing, and at other times properly described one fragment as a unit of analysis. Their inconsistent treatment of statistical analysis erodes scientific robustness of the main results.

Second, I was not able to find any justification that they obtained enough number of samples per fragment to represent an interaction network in each fragmented habitat. Sampling enough number of samples to robustly capture network characteristics has been poised as a crucial problem in interaction network research, particularly when one is to distinguish truly missing links from insufficient sampling effort. This could be easily assessed by plotting the number of retrieved AMF species (or number of interactions) against the number of samples (plant individuals) in each fragment, and then making comparisons based on the same depth of sample coverage. Standardizing sample sizes despite substantial variation in fragment size makes even more concerning about the possibility that most of the rewiring patterns they detected might be the result of potentially unequal sample coverage across fragments.

Reference:

Jordano (2016) Sampling networks of ecological interactions. *Functional Ecology*.

Chao and Jost (2012) Coverage-based rarefaction and extrapolation: standardizing samples by completeness rather than size. *Ecology*

Chase et al. (2018) Embracing scale-dependence to achieve a deeper understanding of biodiversity and its change across communities. *Ecology Letters*

Third, I do not think that some of the statistical analyses actually support their claims. For example, take a look at l.105. Phylogenetically constrained rewiring. I would have to say that Table 2 addresses only indirectly about patterns of interactions shifts as the response variable was set to the number of OTUs (and hence conveying very limited information about changes in species composition). A more direct way to contrast patterns of interaction shift is to repeat the analysis of Table 1 at the family level. I do not understand at all why the author did not take such a more straightforward approach. Also, this section should be proofread carefully as it contains several examples of poor writing.

Minor comments:

l.37. I don't think that the term 'novel interactions' is used in a consistent way throughout. Also, it wasn't very clear how novel interactions relate to the rewiring concept.

l.44. An irrelevant improvement of community robustness does not make much sense; please revise.

l.78. What about the number of samples per fragment?

l.85. Two jargons, forbidden and realized interactions, come out of the blue. Could you please explain them here?

l.140 and onwards. I feel that discussion is very narrowly scoped compared to the issues raised in Introduction.

l.197. Sample size per fragment is crucial information and should be moved to Results section. It sounds like all terminal roots from every sampled plant individual are sampled and pooled, but I don't think this is realistic. Maybe only a subset of terminal roots was sampled per each?

Fig.1 Although the choice of colors and symbol makes it hard to interpret the result, different species seem to display different increasing/decreasing patterns with larger fragment size. If so, then couldn't it be overly simple to fit a single linear function to the all plant species data?

Fig.3 Most networks look too cluttered and almost impossible to recognize any pattern. It is a good idea to use a different configuration with more transparent color nodes to enhance visualization.

Table2. What is the unit of statistical analysis?

Reviewer #2 (Remarks to the Author):

Brief summary of the manuscript

Gypsum patches varying in size were selected, species of plants and AM fungi were identified, and interaction networks analyzed. The authors found that AM networks were more similar between similar than dissimilar-sized patches, and the differences between similar-sized networks were mainly due to rewiring patterns, while differences between dissimilar-sized patches were driven by species turnover.

Overall impression of the work

Assessing network topologies is interesting and could help predict consequences of species loss under different scenarios: habitat fragmentation/degradation, and species invasion. However, more consideration is needed on the underlying biology of the plant-fungal system. The study feels untethered from reality.

Specific comments

1. The language used throughout the ms implies that networks have been perturbed. Specifically, at one time the fragments were larger and connected, and we observe the remnants of that landscape today. The examples in the Intro discuss the loss of species, novel interactions, conversion into impoverished communities, lost partners, etc., all of which imply there has been an event to shift network topology. But the current study does not map on to the conceptual model—these gypsum patches have not shrunk, they just differ in size. This study is not one of cause and effect as the Introduction sets it out to be. I might do some rewording to the change the language to better reflect the layout of the current study. For example, line 53 I would revise from '...provide interesting natural experimental sites for the investigation of the effect of fragmented landscapes on ecological networks' to '...provide survey sites to compare ecological networks across patches differing in size'.

2. More background needs to be provided on the natural system to evaluate the results. What is the matrix surrounding the gypsum outcrops? Is it unsuitable for plants? Is it unsuitable for fungi? If there are plants growing in the matrix, the patchiness of the gypsum may be irrelevant to AMF. Specifically, the gypsum patches may not restrict the distribution of AMF communities in the same way it might restrict plant species. Are plant hosts isolated? Is the current system similar to something like Peay et al. 2010? What is the distance between patches? If the distance is small, AMF may be easily dispersed between patches. How are the patches configured? Is it possible there is some other underlying gradient that is making similar-sized patches more similar in network topology than dissimilar-sized patches? After all, patch size is not replicated and this is a survey.

3. The assumption that plants won't disappear with the loss of their fungal partners, and AMF will is tenuous (lines 57-61). Viable spores can lay dormant in soils, and there is likely dispersal of spores among patches (Bueno and Moora, 2019). So, it depends on how long AMF are without a partner. I am not sure what time scales are assumed in the current study. These gypsum patches are a geological feature in the landscape, so I assume we are dealing with long time scales (i.e., there has been lots of time for dispersal among plants and fungi)? Importantly, at this time scale, how can partners be deemed 'novel'? (e.g. line 151 in the Discussion). The unstated assumption is that the plant community present within a given patch has not changed over long time scales (unlikely), and even if this were possible there

is much overlap in plant species composition among patches, so I can't make out how partners could be novel. The mycorrhizal status of plants will also influence their survival following the loss of AMF. Plants are categorized as obligate, facultative or non-mycorrhizal. For example, information on mycorrhizal status for European plant species is located in the MycoFlor database (Hempel et al. 2013).

4. 1799 unique fungal OTUs for a single study seem awfully high. Were these OTUs monophyletic? For context, Kivlin et al. 2011 conducted a study at the global scale and observed 563 OTUs and 669 OTUs for the 18S and 28S regions, statistically different AM fungal communities only occurred between continents, and within continents, AM fungal communities did not differ significantly with geographic distance (Kivlin et al. 2011). In another global survey, only 236 AMF species were found (Davison et al. 2015). Given the findings of the previous surveys at the global scale, I find it difficult to believe the results presented in the current study. I suspect something funky happened in the bioinformatics. Though the plant species are presented, there were no data for the species of AM fungi. The authors may want to consider including this information in their revisions. As the AMF richness of the site seems vastly inflated, I did not evaluate the results past this point.

5. Random comments

Line 22 : this study is a snapshot of networks, so one might argue the current work is also 'static'.

Line 58: 'therefore the ABSENCE of all the host plants...'

Figure 3 & 4: how sensitive are these results to the selected binning of patch sizes?

Bueno and Moora. 2019. How do arbuscular mycorrhizal fungi travel? *New Phytologist* 222: 645-647.

Davison et al. 2015. Global assessment of arbuscular mycorrhizal fungi diversity reveals very low endemism. *Science* 349: 970-973.

Hempel, S., L. Götzenberger, I. Kühn, S. G. Michalski, M. C. Rillig, M. Zobel, and M. Moora. 2013. Mycorrhizas in the Central European flora—relationships with plant life history traits and ecology. *Ecology* 94:1389–1399

Kivlin, Hawkes and Treseder 2011. Global diversity and distribution of arbuscular mycorrhizal fungi. *Soil Biology and Biochemistry* 43: 2294-2303

Peay, Garbelotto and Bruns. 2010. Evidence of dispersal limitation in soil microorganisms: Isolation reduces species richness on mycorrhizal tree islands. *Ecology* 91: 3631-3640

Reviewer #3 (Remarks to the Author):

The question of how habitat fragmentation affects the interaction structure of mutualisms is compelling, and the chosen study system has the potential to lend significant insight into this question. However, I have significant concerns about the methods of data analysis and conceptual approaches used to make inferences here.

First, in the analysis of how fragment size affects "interaction load" (number of fungal species per plant individual), why was "fragment" not used as a random effect in the mixed model? There were multiple samples per fragment, and as currently analyzed, those samples are treated as independent replicates for each fragment size. It is not clear whether this would affect the results, but the potential non-independence of samples from the same fragment should be modeled.

Perhaps more important, conceptually, I don't understand how the influence of fragment size on "interaction load" can be distinguished from a simple effect of fragment size on AM fungal diversity? If larger fragments have larger AM fungal diversity (as we might expect), wouldn't

they also have higher "interaction load"? So, what is the utility of considering "interaction load" rather than simply asking how fragment size affects AM fungal diversity? How do "interactions" affect these results, if at all? Why is Figure 1 not simply a demonstration of a species-area relationship in AM fungi?

I don't understand the hypothesis that similar-sized fragments should show a low species turnover and high rewiring, while between dissimilar-sized fragments species, turnover might contribute more than rewiring to shifts in interaction patterns. Why should similar-sized fragments have more similar species composition than different-sized fragments? Doesn't this idea depend on the assumption that particular AM fungal species are specialized to either small or large fragments? This hypothesis is not explained, and I don't see how it links to the biology of AM plants or fungi. Finding patterns in these network metrics, without concrete links to the biology of the organisms, is not very satisfying.

lines 331-332: I don't understand this metric for phylogenetic diversity: "the number of OTUs of each fungal family present in each plant species or family." How does this metric represent phylogenetic diversity? The number of fungal families could represent a crude metric of phylogenetic diversity, but that doesn't seem to be what this metric represents. Moreover, phylogenetic diversity is a complex idea, with a fairly large literature devoted to the various ways it can be quantified. As presented, the work here does not take advantage of these various nuanced concepts of phylogenetic diversity.

In addition, testing phylogenetic constraint on "rewiring" using family- vs. species-scale comparisons seems like a coarse approach to this problem--finding a species effect plus a lack of a family effect may indicate that rewiring happens primarily within families (and not between), as suggested by the results, but it lacks precision because "family" is a relatively arbitrary approach to quantifying phylogeny, and the "species" effect is both within and between families. Why not test this idea more precisely, using a phylogenetically-explicit approach, which would be more powerful? Why not analyze rewiring within and between particular clades?

lines 294-296: Why were generalized linear models used here for testing "whether the number of the different types of interactions depended on the fragment size"? I don't understand the description of this analysis--what were the predictor and response variables, and why was a non-Gaussian error distribution used (and what was it)?

The finding of 1799 unique OTUs of AM fungi is surprisingly high, given that it exceeds many global estimates for the number of molecular AM fungal species (see, e.g., Opik et al. 2014, *Botany* 92: 135–147). I am not an expert in AM fungal taxonomy or molecular methods, but this observation raises a red flag for me--why did the authors find such surprising hyper-diversity of AM fungi? Is it an artifact of the methods used? How might the results have been affected by the choice of primers, which involves some significant considerations and trade-offs for AM fungi?

The background justification for the study, specifically regarding the mycorrhizal biology (lines 57-65), was a bit confusing. First, the authors do not specify that they are referring to arbuscular mycorrhiza (rather than, e.g., ectomycorrhizal). Arbuscular mycorrhizal differ from ectomycorrhizal in their patterns of host-specificity, but as written, the authors seem to be making general statements about patterns of specificity in all mycorrhizal interactions. Also, I don't think the Klironomos paper (citation #15) is an appropriate citation for this statement: "mycorrhizal symbiosis has been considered traditionally as a highly generalist interaction (15) recent evidence has shown that these ecological interactions are non-random in natural communities (16)." In fact, the Klironomos paper found evidence for host preference, and doesn't really represent a "traditional" view. Rather, it represents evidence for the emerging view that "preference" is important in these communities.

Some more minor comments follow below:

lines 81-92: The formatting of the t-test results here is a bit confusing: First, why are df given for the first result, but not for the rest? Second, what are the parameter estimates

given in parentheses? Are they slopes?

line 199: insert "per fragment" after "individuals"

line 204: how were the samples dried? what temperature, and over how many hours?

line 373: delete the first instance of "and"

We consider that the new version has substantially improved its clarity, and we are grateful to the reviewers for their contribution to this improvement. In the next lines we summarize and respond to all the major concerns arisen by the three referees, and afterwards we attend their specific comments following a one-by-one points structure.

Main concerns Reviewer 1:

- The authors sometimes analyzed **plant individual as an independent statistical unit for significance testing**, and at **other times properly described one fragment as a unit of analysis**.

Following the reviewer's suggestion, we have removed the analyses performed at the individual plant level (1. Linear models involving mycorrhizal load in the section "dependence of the interactions loss on the fragment size" and 2. Multivariate analyses based on the fungal community present in each plant individual in the section "phylogenetically constrained rewiring").

Specifically, we have performed the following changes:

1. Regarding the section "dependence of the interactions loss on the fragment size":

We have removed all the analyses involving fragment size, including this section, also attending Reviewer's 2 main concern, who stated that: *"**The language used throughout the ms implies that networks have been perturbed. Specifically, at one time the fragments were larger and connected, and we observe the remnants of that landscape today. these gypsum patches have not shrunk, they just differ in size. This study is not one of cause and effect as the Introduction sets it out to be. I might do some rewording to the change the language to better reflect the layout of the current study**"*,

We have re-worded and re-structure the manuscript, and also re-analyzed the data to fully assess this comment. Therefore, we removed those analyses that assessed tendencies along fragments' size, also attending reviewer 3 main comment: *"I don't understand the hypothesis that similar-sized fragments should show a low species turnover and high rewiring, while between dissimilar-sized fragments species, turnover might contribute more than rewiring to shifts in interaction patterns. Finding patterns in these network metrics, without concrete links to the biology of the organisms, is not very satisfying"*. Specifically, we removed the following analyses: 1. Generalized linear models for each interaction type to test whether the number of the different types of interactions depended on the fragment size; and 2. Linear mixed models to test whether β_{WN} , β_{ST} , β_{OS} , and β_{ST}/β_{WN} could be explained by size differences between each pair).

Instead, in the new version we base on a) the comparisons across every possible pair of fragments, (without taking their size into account) to estimate the contribution of rewiring and species turnover to the shifts in interaction patterns across fragments, and b) the frequency with which each plant species-fungal OTU interaction is realized across the 15 fragments to define preferred and non-preferred hosts of each fungal OTU, and test whether the non-preferred hosts with which a fungal OTU interacts (i.e. rewiring) tend to be close relatives (i.e. low phylogenetic distance) of any of the preferred hosts.

2. Regarding the section “phylogenetically constrained rewiring”

We have included in the new version a new approach to assess whether rewiring is phylogenetically conserved using fragments as units of analyses, as suggested by reviewer 1, and also attending a main concern of reviewer 3, who stated: “*the number of OTUs of each fungal family present in each plant species or family.*” **How does this metric represent phylogenetic diversity? [...] In addition, testing phylogenetic constraint on “rewiring” using family- vs. species-scale comparisons seems like a coarse approach to this problem[...]. Why not test this idea more precisely, using a phylogenetically-explicit approach”**

In the new approach, which is described in the new version of the methods section under the heading “phylogenetically constrained rewiring”, we assess whether the non-preferred hosts with which a given fungal OTU actually interacts tend to be close relatives of the preferred hosts. We generated the phylogenetic relationships between all the plant species in our study using the R function S.PHYLOMAKER (Qian & Jin, 2016), which uses the PhytoPhylo backbone megaphylogeny (i.e. an updated version of the time-calibrated angiosperm species-level phylogeny (Zanne et al., 2014)).

Specifically, we used the following approach:

- Using the information across the 15 fragments we quantify how many times each pair of plant species-fungal OTUs occurs in the same fragment, and consider that as the number of potential interactions between them. Then, for each fungal OTU we defined the preferred hosts as the plant species that actually harbor it in more than 50% of their potential interactions (i.e. times that both partners co-occur across fragments). Only cases where the number of potential interactions were 4 or more were taken into account, avoiding extremely high or low stochastic percentages due to a low number of potential interactions. The rest of plant species that also harbored the fungal OTUs were considered non-preferred host.
- For each OTU in each fragment, we calculated the minimum phylogenetic distance between the non-preferred hosts actually harboring the OTU in that fragment, and their respective closest relative among the preferred hosts of the OTU, and then averaged the values for all the non-preferred hosts of the OTU in the fragment.
- Simultaneously, we built a null model by randomly selecting a set of non-preferred hosts that could have potentially harbored the OTU in that fragment. To do so, from the plant species present in the fragment (excluding the preferred hosts for each OTU), we randomly selected the same number of observed non-preferred hosts harboring the OTU, and calculated the same metric described in the previous paragraph. Finally, we tested whether the difference between the observed and expected values, based on the null model, was significantly different from 0 using a t test.
- A justification that they obtained **enough number of samples per fragment to represent an interaction network in each fragmented habitat** is required

As suggested, we have included a new figure in the supplementary information (Fig. S1) showing the accumulated number of OTUs per number of plants in each fragment.

- **Standardizing sample sizes despite substantial variation in fragment size** makes even more concerning about the **possibility** that most of the **rewiring patterns** they detected **might be the result of** potentially **unequal sample coverage across fragments**

We would like to clarify that we did not standardize sample size across fragments, as line-transects were proportional in length to the area of the fragment. This was stated in the methods (section: field sampling). Therefore, the sample coverage across fragments is indeed similar across fragments (i.e. proportional to their size). We have included a remark in the new version to avoid this confusion in the future (“Therefore, despite the variation in fragment size, the sampling ensured a similar proportion of sample coverage across fragments”). However, we consider that sampling the same number of plant individuals in all fragments is necessary indeed, because otherwise, it will have resulted in an unequal sampling of interactions across fragments, preventing a reliable comparison among fragments.

- **A more direct way to contrast patterns of interaction shift is to repeat the analysis of Table 1 at the family level**

We have done the analyses suggested which shows that the Bst/Bwn (i.e. contribution of species turnover relative to rewiring to shifts interaction patterns between networks) is lower (and thus rewiring higher) when the networks are defined at the family level compared to when they are defined at the species level ($t=2.5$, $df = 174$, $p\text{-value} = 0.01$, mean Bst/Bwn at the species level = 0.641, mean Bst/Bwn at the family level = 0.598).

However, this analysis/result has not been included in the new version of the manuscript because we have assessed interaction shifts attending to reviewer’s 3 comment, by considering the phylogenetic distance between preferred and non-preferred hosts instead of using a family- vs. species-scale comparison. Please see our first answer to the reviewer (i.e. 2. Regarding the section “phylogenetically constrained rewiring”) for further details.

Main concerns Reviewer 2:

- **The language** used throughout the ms **implies that networks have been perturbed**. Specifically, at one time the fragments were larger and connected, and we observe the remnants of that landscape today. **these gypsum patches have not shrunk, they just differ in size**. This **study is not one of cause and effect as the Introduction sets it out to be**. I might do some **rewording** to the change the language to better reflect the layout of the current study.

We have fully attended this comment by re-wording, re-structuring the paper and re-analyzing the data. Please see our answer to reviewer 1 (in 1. Regarding the section “dependence of the interactions loss on the fragment size”). In the new version we focus on the main aim of our manuscript (i.e. weather network rewiring tends to be phylogenetically conserved, and its effects on community stability), and removed those analyses that assessed tendencies along fragments’ size, to avoid suggesting the implication the reviewer refers to. In addition, we have revised all the manuscript re-wording the text accordingly.

- More background needs to be provided on the natural system to evaluate the results.

What is the matrix surrounding the gypsum outcrops?

Mainly limestone, this information has been included in the study design section

Is it unsuitable for plants?

Is it unsuitable for extremely gypsophilous plant species which can only establish in gypsum soils. We have clarified the definition of gypsophilous in the new version

Is it unsuitable for fungi?

Unfortunately, there is not enough literature about the gypsophilia in arbuscular mycorrhizal fungi. Although the authors are experts in fungal communities in gypsum soils, the comparison of the fungal communities present in gypsum outcrops and outside gypsum outcrops, close by, has not been explored as far as we know.

If there are plants growing in the matrix, the patchiness of the gypsum may be irrelevant to AMF.

Maybe, but in that case, the AMF communities colonizing all fragments will tend to be similar, so that differences in interaction networks across fragments will only depend on plant species turn over. However, it doesn't seem likely as our results show that the network differences across fragments are similarly explained by rewiring and species turnover (the contribution of species turnover to explain network differences is 64% of the contribution of rewiring (Bst/Bwn; Table S2).

Specifically, the gypsum patches may not restrict the distribution of AMF communities in the same way it might restrict plant species. Are plant hosts isolated? Is the current system similar to something like Peay et al. 2010? What is the distance between patches? If the distance is small, AMF may be easily dispersed between patches. How are the patches configured?

We agree with the reviewer that it will be interesting to know more precisely to which extent gypsum soil restricts the colonization, dispersal and distribution of plant and fungal communities and whether this differentially affects the composition of guilds in each site. However, the purpose of our manuscript was not to obtain a mechanistic understanding of why fragments differ in this system (as the study design does not allow to do so). Instead, analyze the observed differences between the interaction patterns across 15 sites, decompose those differences into species and interactions turn over, and try to identify patterns comparing the interactions that appear more consistently or less frequently across sites, using the latter as a proxy of rewiring.

- The assumption that plants won't disappear with the loss of their fungal partners, and AMF will be tenuous (lines 57-61). Viable spores can lay dormant

in soils, and there is likely dispersal of spores among patches (Bueno and Moora, 2019). So, it depends on how long AMF are without a partner. I am not sure what time scales are assumed in the current study.

In this paper we combine a) the data collection of plant and AMF species, and characterization of the interaction networks in the 15 sites, with b) a simulation exercise based on rewiring rates obtained from the data, in which we compare what would be the response of AMF communities under different scenarios (no rewiring, random rewiring and phylogenetically constrained rewiring) given a set of assumptions. These different scenarios share the same assumptions (i.e. a given AMF would disappear if all their partners are not present in a given location or disregard the AMF species not found in the roots sampled), therefore differences in the outcome of the different scenarios cannot be attributed to the reliability of these assumptions. Thus, our conclusion that interactions rewiring increases the robustness of the networks, even when this rewiring is phylogenetically constrained, cannot be attributed to the reliability of the assumptions, as the three scenarios compared are simulated using the same assumption. These simulations, as any, do not aim to predict or reproduce the exact outcome of communities and ecological interactions in nature. Instead, these simulations theorize about the different responses that a given community will show under different simulated scenarios. This theoretical exercise allows exploring whether the alteration of certain “rules” (in our case allow or not rewiring, and allow it freely or phylogenetically constrained) would result in different outcomes, everything else being equal.

- These gypsum patches are a geological feature in the landscape, so I assume we are dealing with long time scales (i.e., there has been lots of time for dispersal among plants and fungi)? **Importantly, at this time scale, how can partners be deemed ‘novel’?** (e.g. line 151 in the Discussion). The unstated assumption is that the plant community present within a given patch has not changed over long time scales (unlikely), and even if this were possible **there is much overlap in plant species composition among patches, so I can’t make out how partners could be novel.**

We have rephrased the language along the manuscript to avoid terminology that suggests “loss” or “novelty” of interactions. This has been replaced by interactions that consistently appear across sites when the two partners co-occur (i.e. involving preferred hosts) vs. those interactions that most of the times that the two partners co-occur the interaction is not actually realized (i.e. involving non-preferred hosts). We considered that when a given fungal OTU is actually harbored by one of its non-preferred host, this can be used as a proxy of interactions that might have originated by “rewiring”. Please see further details of our changes in this regard in our response to reviewer 1 (1. Regarding the section “dependence of the interactions loss on the fragment size).

- The assumption that plants won’t disappear with the loss of their fungal partners, and AMF will is tenuous (lines 57-61).(information on mycorrhizal status for European plant species is located in the **MycoFlor database (Hempel et al. 2013).**)

We have revised the information available in MycoFlor database and have checked that any of the 28 plant species used in our study are within the 942 plant species classified as obligate mycorrhizal in Hempel et al. 2013. Considering the well-known nature of mycorrhizal fungi as obligate symbionts (i.e. they depend on their plant partners to complete their life cycle), and the lack of evidence that the plant species considered in our study cannot survive without the mycorrhizal fungi, it seems reasonable to invest our first attempt to explore rewiring patterns in this symbiosis from the fungal perspective. Nevertheless, it will be interesting to perform future research considering systems with evidence of mutual dependency to expand our knowledge about interactions rewiring patterns.

- 1799 unique fungal OTUs for a single study seem awfully high. **Though the plant species are presented, there were no data for the species of AM fungi.**

We appreciate this comment. There was a mistake in the original version, and 1799 refers to the number of unique sequences obtained. When the unique sequences were grouped in operational taxonomic units (OTU), it resulted in 161 OTUs. We have included a new sheet in Table S1 of the supplementary material, with the information of fungal OTU in each location.

Main concerns Reviewer 3:

- **First, in the analysis of how fragment size affects "interaction load" (number of fungal species per plant individual), why was "fragment" not used as a random effect in the mixed model?**

As explained above, we have removed all the analyses involving fragments' size, including this one, attending to previous comments of the reviewers. Please see our answer to reviewer 1 (in 1. Regarding the section "dependence of the interactions loss on the fragment size") for further details.

Nevertheless, we have re-done the original analyses including fragment as a random effect (just to attend the reviewer comment, but not included in the new version) and the results were the same:

- **I don't understand how the influence of fragment size on "interaction load" can be distinguished from a simple effect of fragment size on AM fungal diversity?. How do "interactions" affect these results, if at all? Why is Figure 1 not simply a demonstration of a species-area relationship in AM fungi?**

The analyses involving "mycorrhizal load" and fragment size have been removed in the new version of the manuscript, as explained in the previous comments to reviewer 3 and first response to reviewer 1 (Regarding the section "dependence of the interactions loss on the fragment size"). Please see further explanation for this removal in our answer to reviewer 1.

- **I don't understand the hypothesis that similar-sized fragments should show a low species turnover and high rewiring, while between dissimilar-sized fragments species, turnover might contribute more than rewiring to shifts in interaction patterns. Finding patterns in these network metrics, without concrete links to the biology of the organisms, is not very satisfying.**

This part of the manuscript (involving results based on fragment-size) has been removed as explained in the previous comments to reviewer 3. Please see our answer to reviewer 1 (in 1. Regarding the section “dependence of the interactions loss on the fragment size”) for further details.

- I don't understand this metric for phylogenetic diversity: "the number of OTUs of each fungal family present in each plant species or family." **How does this metric represent phylogenetic diversity? The number of fungal families could represent a crude metric of phylogenetic diversity, but that doesn't seem to be what this metric represents.** Moreover, phylogenetic diversity is a complex idea, with a fairly large literature devoted to the various ways it can be quantified. As presented, the work here does not take advantage of these various nuanced concepts of phylogenetic diversity.

We have replaced the previous approach comparing family vs. species level results by a new one considering specifically phylogenetic distance, and therefore avoiding the use of taxonomic classifications. We generated the phylogenetic relationships between all the plant species in our study using the R function S.PHYLOMAKER, and assess whether the non-preferred hosts actually harboring a given fungal OTU tend to involve close relatives to the preferred hosts of fungal OTU. Please see our answer to reviewer 1 (in 2. Regarding the section “phylogenetically constrained rewiring”) for further details

- In addition, **testing phylogenetic constraint on "rewiring" using family- vs. species-scale comparisons seems like a coarse approach to this problem-** finding a species effect plus a lack of a family affect may indicate that rewiring happens primarily within families (and not between), as suggested by the results, but it lacks precision because **"family" is a relatively arbitrary approach to quantifying phylogeny, and the "species" effect is both within and between families.** Why not test this idea more precisely, using a **phylogenetically-explicit approach**, which would be more powerful? Why not analyze rewiring within and between particular clades?

We have fully attended this comment considering specifically phylogenetic distance instead of family-vs. species-scale comparisons, as suggested. Please see our previous answer and also our answer to reviewer 1 (in 2. Regarding the section “phylogenetically constrained rewiring”) for further details

- lines 294-296: **Why were generalized linear models used here for testing** "whether the number of the different types of interactions depended on the fragment size"? I don't understand the description of this analysis--what were the predictor and response variables, and why was a non-Gaussian error distribution used (and what was it)?

This analyses, as any other involving fragment-size have been removed in the new version of the manuscript, in order to attend the reviewer's 2 suggestion regarding the avoidance of the use of fragment size as a proxy for “shrinkage”. Please see our answer to reviewer 1 (1. Regarding the section “dependence of the interactions loss on the fragment size”) for further details.

- **The finding of 1799 unique OTUs of AM fungi is surprisingly high.**

As explained in the previous answer to reviewer 2, there was a mistake in the original version, and 1799 refers to the number of unique sequences obtained. When the unique sequences were grouped in operational taxonomic units (OTU), it resulted in 161 OTUs. The information about which OTU were present in each fragment has been now included in Table S1.

- The background justification for the study, specifically regarding the mycorrhizal biology (lines 57-65), was a bit confusing. **First, the authors do not specify that they are referring to arbuscular mycorrhiza.**

We have specified in the new version that we are referring to arbuscular mycorrhiza

- Also, I don't think the **Klironomos paper** (citation #15) is an appropriate citation for this statement: "mycorrhizal symbiosis has been considered traditionally as a highly generalist interaction (15) recent evidence has shown that these ecological interactions are non-random in natural communities (16)). In fact, the Klironomos paper found evidence for host preference, and doesn't really represent a "traditional" view "

Attending the reviewer's suggestion, we have rephrased the sentence to clarify that that paper doesn't represent a "traditional view". We based on this part of the abstract of Klironomos' paper, to state that this paper supports that "arbuscular" mycorrhizal symbiosis has been considered a generalist interaction: "Arbuscular mycorrhizal (AM) fungi are mutualistic symbiotic associations between 150 species of fungi and the roots of approximately 300 000 species of plants. As a result of this low fungus:host species ratio, it has been assumed that the fungi are not host-specific and that there is a high functional redundancy among fungal species. In this study, fungi and plants were isolated from an old field meadow and used in a series of experiments to test for host-specificity and functional redundancy. AM fungi were not host specific - 96% of combinations yielded successful colonization.", although they show differential responses in P uptake, protection against non-mycorrhizal fungal infection, and productivity for almost every plant-fungus combination tested.

Minor comments:

Reviewer #1

I.37. I don't think that **the term 'novel interactions'** is used in a consistent way throughout. Also, it wasn't very clear how novel interactions relate to the rewiring concept.

We have avoided the term novel in the new version of the manuscript, and have clarified from the abstract the link with rewiring: "Among all potential interactions (i.e. when the two potential partners occur in the same fragment), the interactions that were consistently realized across fragments were considered to involve preferred hosts. On the contrary, interactions that were not realized most of the times that both potential partners co-occur in the same fragment were considered to involve non-preferred host. We used the actual presence of an interaction with non-preferred hosts as a proxy for rewired interactions."

I.44. An irrelevant improvement of community robustness does not make much sense; please revise.

“Irrelevant” has been changed by “low”

I.78. What about the number of samples per fragment?

That information was specified in the methods: “We sampled 15 individuals per fragment for a total of 225 individuals belonging to 28 plant species”

I.85. Two jargons, forbidden and realized interactions, come out of the blue. Could you please explain them here?

That part of the manuscript (involving relationships between interactions types and fragment-size) has been removed in the new version

I.140 and onwards. I feel that discussion is very narrowly scoped compared to the issues raised in Introduction.

In the new version, we have tried to focus more on the importance of assessing potential patterns of interactions rewiring both in the introduction and the discussion.

I.197. Sample size per fragment is crucial information and should be moved to Results section. It sounds like all terminal roots from every sampled plant individual are sampled and pooled, but I don't think this is realistic. Maybe only a subset of terminal roots was sampled per each?

Yes, not all terminal roots were sampled, 500mg of terminal roots of each plant was sampled, mixed and used to characterize fungal symbionts in each plant. We have clarified it in the new version.

Fig.1 Although the choice of colors and symbol makes it hard to interpret the result, different species seem to display different increasing/decreasing patterns with larger fragment size. If so, then couldn't it be overly simple to fit a single linear function to the all plant species data?

This figure and analyses have been removed as the rest of analyses involving fragment-size, please see answer to reviewer 1 (1. Regarding the section “dependence of the interactions loss on the fragment size) for further details.

Fig.3 Most networks look too cluttered and almost impossible to recognize any pattern. It is a good idea to use a different configuration with more transparent color nodes to enhance visualization.

This figure has been removed. Please see our answer to reviewer 1 for further details about the main changes in the new version.

Table2. What is the unit of statistical analysis?

The units were plant individuals, but this analyses have been removed attending to the comments of reviewer 1. Please see our answer to reviewer 1 (1. Regarding the section “dependence of the interactions loss on the fragment size) for further details.

Reviewer #2

Line 22 : this study is a snapshot of networks, so one might argue the current work is also 'static'.

We agree that our data is a snapshot of networks, but we consider that our approach, i.e. comparing the different snapshots, provides a different perspective than a single snapshot, as it allows the assessment of variation across interaction networks.

Line 58: 'therefore the ABSENCE of all the host plants...'

"Loss" has been replaced by "absence"

Figure 3 & 4: how sensitive are these results to the selected binning of patch sizes?

The results were obtained using all the possible comparisons across fragments, so there was not any binning in the analyses. The binning was only used to produce the figure, just for illustrative purposes, as indicated in the legend. Nevertheless, these analyses, as all the others involving fragment size, have been removed in the new version of the manuscript

Bueno and Moora. 2019. How do arbuscular mycorrhizal fungi travel? *New Phytologist* 222: 645-647.

Davison et al. 2015. Global assessment of arbuscular mycorrhizal fungi diversity reveals very low endemism. *Science* 349: 970-973.

Hempel, S., L. Götzenberger, I. Kühn, S. G. Michalski, M. C. Rillig, M. Zobel, and M. Moora. 2013. Mycorrhizas in the Central European flora—relationships with plant life history traits and ecology. *Ecology* 94:1389–1399

Kivlin, Hawkes and Treseder 2011. Global diversity and distribution of arbuscular mycorrhizal fungi. *Soil Biology and Biochemistry* 43: 2294-2303

Peay, Garbelotto and Bruns. 2010. Evidence of dispersal limitation in soil microorganisms: Isolation reduces species richness on mycorrhizal tree islands. *Ecology* 91: 3631-3640

Reviewer #3 (Remarks to the Author):

Some more minor comments follow below:

lines 81-92: The formatting of the t-test results here is a bit confusing: First, why are df given for the first result, but not for the rest? Second, what are the parameter estimates given in parentheses? Are they slopes?

These analyses, as all the rest involving fragment size have been removed in the new version

line 199: insert "per fragment" after "individuals"

Changed, thanks

line 204: how were the samples dried? what temperature, and over how many hours?

They were dried with filter paper, but they were not put in the oven

line 373: delete the first instance of "and"

Done, thanks

Reviewers' comments:

Reviewer #1 (Remarks to the Author):

I see the manuscript improved to some extent, but the revision brings new concerns about the statistical analysis. I feel that Results and Discussion are not well developed and still need substantial revision.

Major comments:

I see no coherent story was put forward after revision by integrating a new series of statistical analysis. Notably, it remains unclear how 'host preference' is linked with patterns of 'rewiring' due to the absence of statistical analysis that connects the two concepts together. I suppose that one can test that the observed rewiring occurred in a more phylogenetically constrained set of plant species than expected by chance in any possible pair of fragments, without defining preferred/non-preferred host. That is, the authors could have performed decomposition of network beta diversity first, and by calculating the rewiring beta_{wn} and extracting subweb (composed of a set of species shared between the two fragments), it is possible to get some coefficient of deviation from phylogenetically random rewiring, using a similar approach to calculating beta null deviation (see Jonathan Chase's paper in *Ecosphere* 2011 and Caroline Tucker's paper in *Oikos* 2016). The degree of host preference could also be more explicitly accounted for by performing abundance-based permutation as commonly done in null model analysis. Making a histogram of the beta null deviation in interaction networks appears to be the most straightforward way to assess if rewiring is phylogenetically constrained across the fragmented patches. Given that, I don't think that the current statistical analyses are appropriate and directly test the hypotheses raised in Introduction. Technically, I see t-test applied to bimodal data, undoubtedly violating the assumption of t-test.

Two limitations must be addressed in Discussion. First, this study did not assess rewiring directly as there are no actual temporal observations of networks being rewired. Instead, the authors took an alternative approach of space-by-time substitution using snapshot of many "replicate" fragments in the region. This approach is of course limited by unmeasured environmental heterogeneity of fragment patches that can affect network dissimilarity, leaving the possibility still open that spatial dissimilarity predicts temporal dissimilarity. Second, phylogenetic tree used in the study is a surrogate tree and no original sequencing was performed. Hence the authors may state that the results of phylogenetically constrained rewiring should be interpreted with caution.

Minor comments:

L34. the relative to what?

L39. evolutionarily

L53. such as

L68. I don't understand what is meant by saying 'non-consistent'.

L99. A new paragraph should come from 'Across'

L153. To ensure reproducibility of the study, the authors may provide a georeferenced map as a new Figure, where all the 15 fragments are shown.

L257. Is this partitioning really additive? The original paper by Poisot says otherwise.

L288-291. I don't understand why this metric informs phylogenetically constrained rewiring in the absence of clear linkage of host preference to patterns of rewiring.

Supplementary Figure 1. Could you please show confidence bands as well?

Reviewer #3 (Remarks to the Author):

I have read the revised manuscript, and find that the authors have done an excellent job addressing my substantial concerns regarding the original version of the manuscript. The current version is much improved, more concise and focused, and the new analyses using phylogenetic distance are a significant improvement. The results from these new analyses are exciting, and should be interesting to a broad audience of scientists studying

interactions, networks, symbioses, and global change. I have no remaining significant concerns.

Minor suggestions:

Lines 38-39: change "evolutionary" to "evolutionarily"

Line 53: insert "as" between "such" and "gypsum"

Line 66: change "networks" to "network"

Line 87: change "result" to "results"

Lines 115 and 117: change "what" to "which"

Line 122: change "elucidate" to "elucidated"

Please find below the detailed explanation of all the changes made in the new version of the manuscript in order to attend all the reviewers' comments.

Reviewers' comments:

Reviewer #1 (Remarks to the Author):

I see the manuscript improved to some extent, but the revision brings new concerns about the statistical analysis. I feel that Results and Discussion are not well developed and still need substantial revision.

Major comments:

I see no coherent story was put forward after revision by integrating a new series of statistical analysis. Notably, it remains unclear how 'host preference' is linked with patterns of 'rewiring' due to the absence of statistical analysis that connects the two concepts together.

Our link between "host preference" and "rewiring" is not statistical, but conceptual. We have tried to explain this better adding the following information in the last paragraph of the introduction: Rewiring refers to changes in the configuration of a network through shifts in the interactions that shape the network (i.e. links). The characterization of networks with similar nodes (i.e. fragments with similar species composition) provides a useful study system to maximize the probability of finding multiple pairs of co-occurring species that might interact in some fragments but might not interact in others. This allows estimating of the frequency with which each interaction is actually realized across fragments, as long as both potential partners co-occur. This approach provides a useful proxy of how likely a given interaction is to occur (or not) between two species across fragments. The interactions that are likely to occur, whenever the two partners co-occur, are assumed to involve preferred hosts, while the interactions that are unlikely to occur, even when both partners co-occur in the same fragment, are considered to involve non-preferred hosts. In this sense, it can be tested whether rewiring (i.e. the establishment of interactions with non-preferred hosts) tend to occur with close relatives of the preferred hosts.

I suppose that one can test that the observed rewiring occurred in a more phylogenetically constrained set of plant species than expected by chance in any possible pair of fragments, without defining preferred/non-preferred host.

Regardless of whether each possible pair of fragments is compared instead of comparing each fragment with the rest of fragments together, it is necessary to define rewiring in order to test any pattern of rewiring. In order to define what is (or is not) a rewired link we need to classify interactions into those involving preferred (thus, no rewired interactions) and non-preferred hosts (rewired interactions) (or shared vs non-shared between each possible pair of fragments), so using a pair-wise approach would

not avoid the necessity to define preferred/non-preferred (or shared vs non shared) hosts. We hope our previous explanation has improved the clarity of this idea.

That is, the authors could have performed decomposition of network beta diversity first, and by calculating the rewiring β_{wn} and extracting subweb (composed of a set of species shared between the two fragments), it is possible to get some coefficient of deviation from phylogenetically random rewiring, using a similar approach to calculating beta null deviation (see Jonathan Chase's paper in *Ecosphere* 2011 and Caroline Tucker's paper in *Oikos* 2016).

Neither of the papers proposed provide a way to “get some coefficient of deviation from phylogenetically random rewiring”

We could get a coefficient of deviation from a null model, but neither Chase or Tucker's paper provide a way to test for any phylogenetic patterns in the null models that they use. Our approach instead, specifically allows calculating the phylogenetic distance between the non-preferred hosts actually harboring the fungi in a given fragment and the preferred hosts, and compare it with the phylogenetic distance between the latter and any of the potential hosts present in the fragment.

The degree of host preference could also be more explicitly accounted for by performing abundance-based permutation as commonly done in null model analysis.

Our estimate of host preference also accounts for abundance as it is calculated based on the number of times that the two partners co-occur (i.e. number of potential interactions) across fragments. Therefore, more abundant plant species will be more likely to co-occur with OTUs across fragments, and thus the number of times that the interaction is realized among them must be higher in order to be considered as preferred hosts, compared to other species with a lower abundance. Also notice that, as stated in the methods' section “Phylogenetically constrained rewiring”, “Only cases where the number of potential interactions were 4 or more were taken into account, avoiding extremely high or low stochastic percentages due to a low number of potential interactions”

Making a histogram of the beta null deviation in interaction networks appears to be the most straightforward way to assess if rewiring is phylogenetically constrained across the fragmented patches.

According to the papers proposed, the beta null deviation is used to test whether the B-diversity of species differs from a null model. However, applying that approach to the b-diversity of interactions might not be straight forward, as non-shared links could be due to both shifts of the interactions or species turnover. Instead, Poisot et al. 2012 focused specifically in implementing a way of estimating b-diversity of interactions, instead of species, and therefore, we consider that Poisot's et. al. 2012 is more suitable for our purpose. In addition, as explained above, the proposed papers do not specify how to include the phylogenetic relationships in their approach.

Given that, I don't think that the current statistical analyses are appropriate and directly test the hypotheses raised in Introduction.

We hope the more detailed explanation provided regarding the link between “non-preferred hosts” and “rewiring” helps the readers to better understand why our statistical analyses are suitable for testing the hypotheses raised in the introduction

Technically, I see t-test applied to bimodal data, undoubtedly violating the assumption of t-test.

The referee might refer to Fig 2, but notice that in the density plot the values are not paired. However, the t-test is performed with the “difference between the observed and expected values”, as stated in the methods section “Phylogenetically constrained rewiring”.

Two limitations must be addressed in Discussion. First, this study did not assess rewiring directly as there are no actual temporal observations of networks being rewired. Instead, the authors took an alternative approach of space-by-time substitution using snapshot of many “replicate” fragments in the region. This approach is of course limited by unmeasured environmental heterogeneity of fragment patches that can affect network dissimilarity, leaving the possibility still open that spatial dissimilarity predicts temporal dissimilarity.

We have addressed this limitation in the discussion:

“Nevertheless, these results should be interpreted with caution as due to the difficulty to assess networks rewiring in field conditions, our results are based on an approach of space-by-time substitution, using snapshot of many fragments in the region. Ideally, host preference could be assessed experimentally, exposing fungi to every potential host under similar conditions, and quantifying the fungal association with each plant species relative to the rest, and even monitoring this process over time. However, many of the fungal species found in this study might be difficult to cultivate, what prevents using them in experimental designs, and furthermore, conducting these experiments in a broad range of plant species under field conditions might easily become unfeasible.”

Second, phylogenetic tree used in the study is a surrogate tree and no original sequencing was performed. Hence the authors may state that the results of phylogenetically constrained rewiring should be interpreted with caution.

We have incorporated this idea in the new version of the manuscript,

“Also, the phylogenetic distances between the plant species have not been established by genotyping plant material, and therefore the estimates of the phylogenetic distances between species of the same family might be slightly rough. Instead, we have used the well established relationships at the family level. However, considering that many of the families used are monospecific, and the highest number of species within a family is low (i.e. 4), our approach is likely to provide an acceptable estimate of the phylogenetic distances between the plant species used in this study. Thus, considering the interest and

difficulty to assess rewiring patterns in the wild, our attempt provides valuable information that could guide further necessary investigation.”

Minor comments:

L34. the relative to what?

The original sentence was: “Despite the increasing attention that rewiring has received, we still lack a deep understanding of its relative contribution to the diversity of interactions across communities, the potential rules governing the rewiring of interactions, or its implications for the stability of natural communities”

It has been rephrased as:

“Despite the increasing attention that rewiring has received, we still lack a deep understanding of its contribution to the diversity of interactions across communities, the potential rules governing the rewiring of interactions, or its implications for the stability of natural communities”

L39. Evolutionarily

Done, thanks

L53. such as

Done, thanks

L68. I don't understand what is meant by saying ‘non-consistent’.

We have removed the term “non-consistent”.

The original sentence was “We hypothesize that networks rewiring can be phylogenetically constrained, so that non-consistent interactions with non-preferred hosts tend to occur with close relatives of the preferred hosts”

And now has been rephrased as

“We hypothesize that networks rewiring can be phylogenetically constrained, so that interactions with non-preferred hosts tend to occur with close relatives of the preferred hosts”

L99. A new paragraph should come from ‘Across’

Done

L153. To ensure reproducibility of the study, the authors may provide a georeferenced map as a new Figure, where all the 15 fragments are shown.

The required figure has been provided as supplementary Figure S1.

L.257. Is this partitioning really additive? The original paper by Poisot says otherwise.

Here is the text directly extracted from Poisot et al. 2002 in which they state that the partition is additive:

“THE DISSIMILARITY OF NETWORKS

Additive partitioning

Table 1 synthesizes our partitioning of diversity. Differences in interactions between networks (bWN) originate from differences in species composition (bST, dissimilarity in interaction structure introduced by dissimilarity in species composition), and because shared species between the two realisations may interact differently (bOS,

dissimilarity of interactions in co-occurring species). **This leads to an additive view of network dissimilarity, wherein:**

$$bWN = bST + bOS$$

By definition, bWN and bST, but not bOS, will covary with the species composition dissimilarity between networks (bS). Given that bOS (dissimilarity of interactions between shared species) is a component of bWN, the inequality $bOS \leq bWN$ is always satisfied, and bST takes values between 0 (dissimilarity between two networks is entirely explained by shared species interacting differently), and bWN (the shared species interact in the same way, and all the difference between the two networks is explained by species turnover). Because differences in network structure can arise either through changes in species compositions or realised interactions, there is no obvious analytical solution for bST, which is found by removing the impact of dissimilarity of interactions on the total dissimilarity between networks as indicated above.”

L288-291. I don't understand why this metric informs phylogenetically constrained rewiring in the absence of clear linkage of host preference to patterns of rewiring.

We hope that the detailed information provided in our first answer to the major concerns of the reviewer also helps clarifying this comment.

Supplementary Figure 1. Could you please show confidence bands as well?

The values reported in Figure S1 are raw data (i.e. the accumulated number of fungal OTUs observed per plant individual). Our purpose was not to apply any model not to predict an appropriate sample size, but instead we intended to show the sampling cover obtained. Therefore, no estimate or measurement of deviation of that potential estimate has been calculated as we have preferred to present the raw data in the figure.

Reviewer #3 (Remarks to the Author):

I have read the revised manuscript, and find that the authors have done an excellent job addressing my substantial concerns regarding the original version of the manuscript. The current version is much improved, more concise and focused, and the new analyses using phylogenetic distance are a significant improvement. The results from these new analyses are exciting, and should be interesting to a broad audience of scientists studying interactions, networks, symbioses, and global change. I have no remaining significant concerns.

Minor suggestions:

Lines 38-39: change "evolutionary" to "evolutionarily"

Done, thanks

Line 53: insert "as" between "such" and "gypsum"

Done, thanks

Line 66: change "networks" to "network"

We have changed “network” to “networks”, which we think is what the reviewer meant

Line 87: change "result" to "results"

Done, thanks

Lines 115 and 117: change "what" to "which"

Done

Line 122: change "elucidate" to "elucidated"

Done, thanks

REVIEWERS' COMMENTS:

Reviewer #1 (Remarks to the Author):

The authors did a great job addressing host preference and its link to network rewiring. I have several minor comments on the manuscript.

L101-102. 'Across the 15 networks...' should come as the second sentence of the paragraph. And 'Fungus DNA was'... may be deleted as this detail does not improve reader's understanding of the main story.

L112. It would help readers to understand the metric better with an explicit equation.

L381. Could you please emphasize here that the results are qualitatively robust to changing definition of a preferred host. For example, "we used an arbitrary threshold of 50% to define a preferred host (see L377) and changing the threshold to say 60% (or 70%?) did not alter the conclusion about phylogenetically constrained rewiring" if this is the case. With those additional results as Supplementary Figures, the main conclusion will become more convincing because the probability wasn't highly significant (0.04).

Fig.2 Could you please describe N (number of data points) of the density plot in the legend? By the way, what is driving this bimodal pattern? I guess that some fragments were more likely to reveal rewiring in a phylogenetically constrained way than the other fragments. Or, some OTUs were more likely to exhibit phylogenetically constrained rewiring than the other OTUs? If it is the case, then it may be very interesting to drill down to the underlying biological mechanism of bimodality.

REVIEWERS' COMMENTS:

Reviewer #1 (Remarks to the Author):

The authors did a great job addressing host preference and its link to network rewiring. I have several minor comments on the manuscript.

L101-102. 'Across the 15 networks...' should come as the second sentence of the paragraph. And 'Fungus DNA was'... may be deleted as this detail does not improve reader's understanding of the main story.

Done

L112. It would help readers to understand the metric better with an explicit equation.

The metric has been described more clearly: "the area below the extinction curve described by the number of fungal OTUs that disappear in each sequential host species simulated removal"

L381. Could you please emphasize here that the results are qualitatively robust to changing definition of a preferred host. For example, "we used an arbitrary threshold of 50% to define a preferred host (see L377) and changing the threshold to say 60% (or 70%?) did not alter the conclusion about phylogenetically constrained rewiring" if this is the case. With those additional results as Supplementary Figures, the main conclusion will become more convincing because the probability wasn't highly significant (0.04).

We see the point of the reviewer. However, it should be considered that the hosts of each OTU are defined as either preferred or non-preferred. Therefore, increasing the threshold to obtain a conservative definition of one type of hosts (i.e. 60-70% to define preferred hosts) will result in a poor definition of the other type of hosts (i.e. interactions with a frequency of 59%-69% will be considered as non-preferred). Thus a threshold of >50% to define preferred and <50% to define non-preferred seems the most reasonable for this purpose.

Fig.2 Could you please describe N (number of data points) of the density plot in the legend?

In the legend of Fig. 2 it has been included the sample size (number of OTUs across fragments for which it was possible to calculate the phylogenetic distance between its non-preferred and preferred hosts, N =388).

By the way, what is driving this bimodal pattern? I guess that some fragments were more likely to reveal rewiring in a phylogenetically constrained way than the other fragments. Or, some OTUs were more likely to exhibit phylogenetically constrained rewiring than the other OTUs? If it is the case, then it may be very interesting to drill down to the underlying biological mechanism of bimodality.

As suggested by the reviewer there are variation across fragments and OTUs in the average minimum phylogenetic distance between the non-preferred and preferred hosts:

The main question in our manuscript is to assess whether the observed pattern of rewiring differs from that expected under a non-phylogenetically constrained rewiring. Understanding the mechanisms underlying potential differences between fungal OTUs constrains to shift between host would be very interesting, especially if there was available information about OTU's functional traits that could allow proposing testable hypotheses. Unfortunately, it is beyond the scope of this manuscript to explore these interesting topics, but it will be interesting for further research.